# OpenReview forum: "Direct Acquisition Optimization for Low-Budget Active Learning"
_ICLR.cc/2025/Conference — ICLR 2025 Conference Withdrawn Submission_

### Official Review · Reviewer_zCD8 · 2024-10-30

**Soundness:** 2
**Presentation:** 3
**Contribution:** 2
**Rating:** 3
**Confidence:** 5

**Summary:**

In this paper, the authors present a new Active Learning framework under the low budget setting. The proposed Direct Acquisition Optimization (DAO) algorithm uses the influence function to update the model parameter and estimate the true loss with importance sampling. The main contributions of this paper is a novel AL algorithm with the proposal of selecting samples based on expected loss reduction. Compared to existing approaches, this method do not rely on additional holdout labeled data for loss evaluation and requires less time for model update using influence function approximation. The experiment results indicate the proposed method is more efficient than the baseline methods including the random selection in image classification use cases.

**Strengths:**

Originality:
This paper focus on extreme low budge setting while using expected loss reduction as criterion for sample selection. Most of active learning are based on heuristics, which emphasize the computational efficiency in sample selection process. But the training objective is different from the heuristic for sample selection, making them less effective. However, evaluating expected loss on test data requires some labeled validation set. This is not suitable for active learning with low budget. For this reason, this paper raised an important challenge in active learning. This paper combines several existing techniques such the the use of surrogate proposal (which is common for sampling-based methods), and the use of variance-reduction importance sampling for estimation of expected loss. It is interesting to see that the influence function is chosen.

Quality:
Both derivations and results from this paper seem correct, and are supportive to the author's claim that the heuristic-based methods are sub-optimal and using surrogate function would help to mitigate the lack of labeled samples in low budget setting. No code is shared thus we cannot judge on repeatablilty of experiment.

Clarity:
This paper has well-organized structure. The objective is clear for the active learning.

**Weaknesses:**

Contribution:
This paper focus on active learning with low budge setting and the algorithm presented solves three issues: 1) how to obtain pseudo-labels  in unlabeled data; 2) how to approximate new model parameter without retraining; 3) how provide low bias estimation on true expected loss on unlabled data.

The issue for 1) is that the obtaining a good proposed surrogate function is difficult.This model plays a central role for the success of this algorithm both theoretically and practically showing in experiments. It is always possible to use the old model from last iteration for surrogate, but it is an poor estimate of the oracle given low budget setup. Thus although this framework claims that it is superior to the heuristic it still inject the heuristic within the surrogate model. More issue with surrogate model is the update of the model. Would you retrain the surrogate model and the main model using the same acquired labels? how to make sure that surrogate model guides us towards the true target better than current production? if not, the cross entropy correction would be a misleading term.
- BTW, the deep learning score is also used as confidence when computing the parameter update. It should be aware that the possible overconfidence issue for deep learning would be a factor to watch esp. when the data is noisy.

The issue for 2) is that to approximate model parameter using influence function, we have to approximate the influence function itself with some iterative method. Both CG and Quasi-Newton like solution has its own challenge, e.g. in CG, in order to compute the influence of one unlabeled sample. we have to do two back-propagation, which is more time-consuming than retraining the model with early stopping. Another issue for this approximation method is additional memory requirement for large model, like LLM. Finally, when the Hessian is indefinite ( e.g. saddle points), this is not clear that if this second-order estimate would result in severe underwhelming result in parameter estimation. Since the new model parameter is used in cross-entropy correction term in importance sampling, it is likely that the error rate will affect the sample stage.

The issue for 3) is that since cross-entropy acquisition function is the key for bias-variance reduction, it is critical to have some guarantee that the impact of error in both surrogate function selection, and parameter update is bounded. It is expected that when the cross-entropy is low but both proposal and target function are limited, this active learning would be worse than random selection.

All in all, a major concern for this model is that it is a very complicated system with multiple approximation steps, which has its own errors in each step. The good performance is thus not guaranteed. The uninformative choice of surrogate function and the approximation of parameter update would damage the performance of this model in larger and more complicated dataset. As for the computation cost as compared to model retraiin, it is not clear parameter update with second-order gradient like estimation would reduce the cost of computation as compared to partial retrain (like using Contrastive Divergence).

**Questions:**

- page 6, Algorithm 1 description, line 5 "randomlly sample n_ivp from unlabeled data U_i,t" this is wrong from the description in page 4, 183 line. We should sample labeled data instead of unlabeled data to update the model parameter (see line 227) But U_i,t from the description is the unlabeled data

- how the label likelihood from surrogate is inferred? did you use the softmax score?

- what is the memory complexity for computing the influence function?

- what is the lift of performance from the random re-ranking in batch acquisition in 3.5 ?

To address issue 1) related questions:
1. The criteria used for selecting and updating the surrogate model?
2. How the authors ensure the surrogate model remains a good estimate of the oracle in low-budget settings?
3. How the quality of the surrogate model affects the cross-entropy correction term?

To address issue 2) related questions:
1. Could author provide a detailed computational complexity analysis comparing DAO to model retraining, especially for larger models?
2. Could author provide empirical evidence comparing the runtime of their method to retraining with early stopping?
3. How DAO handles cases when the Hessian in influence function computation is indefinite? How the method performance near saddle points?
4. Could authors provide an analysis of how errors in parameter estimation propagate to the sample selection stage?

To address issue 3) related questions:
1. It will be nicer if we can see experiments to empirically demonstrate the method's robustness (or lack thereof) to errors in the surrogate function and parameter updates.
2. May need an ablation study showing the impact of each approximation step on the overall performance.

---

### Official Review · Reviewer_cMbQ · 2024-11-04

**Soundness:** 3
**Presentation:** 2
**Contribution:** 2
**Rating:** 5
**Confidence:** 3

**Summary:**

The paper proposes DAO, an expected error reduction strategy for deep active learning. The method leverages influence functions in approximating the model update when training on a newly labeled data point. To further make this approximation computationally efficient, the authors propose to use conjugate gradients and stochastic estimation. In addition, the algorithm also utilizes a low variance approximation of the error rate on the unlabeled pool, utilizing a surrogate model.

**Strengths:**

The paper conducts experiment across a large number of datasets, showing the consistent improvement in accuracy over baseline methods.

The paper also draws from multiple latest advances in deep active learning, and effectively integrated them together to ensure better performance and theoretical justification.

**Weaknesses:**

1. The paper does not compare against TypiClust [1], which is known to outperform all the baseline strategies here in low budget settings.
2. In the high budget experiments, the total budget is not sufficiently large, since each class only has 50 annotation budget on average. It would be interesting to see the algorithm's performance with even larger annotation budget.
3. The notation $\ell(x, f)$ is very confusing. The loss function should be a function of the label $y$ as well. This seems to have been omitted in 3.3 and 3.4. In section 3.3 equation (4), it seems this is approximated using the model's prediction. However, there is no mention of this in section 3.4. How is the loss computed for equations (6) and (9)?
4. If the author is already using the predicted label as ground truth y, instead of the method in section 3.4, the authors can use this to approximate $L_{\text{true}}$, and an ablation study should be conducted. In addition, the authors mention Mussmann et. al. as an expected error reduction algorithm, but never compares against it in the experiments.
5. There has also been quite a few works utilizing NTK to approximate the model's behavior when training on a new data point. The authors should distinguish from these methods, potentially by comparing against them.
6. The authors could consider rearranging the order between sections 3.2 and 3.3. This is because the current 3.2 naturally flows into 3.4, but is interrupted by 3.3 at the moment.

[1] Hacohen, Guy, Avihu Dekel, and Daphna Weinshall. "Active learning on a budget: Opposite strategies suit high and low budgets." arXiv preprint arXiv:2202.02794 (2022).

[2] Mohamadi, M. A., Bae, W., & Sutherland, D. J. (2022). Making look-ahead active learning strategies feasible with neural tangent kernels. Advances in Neural Information Processing Systems, 35, 12542-12553.

[3] Wang, H., Huang, W., Wu, Z., Tong, H., Margenot, A. J., & He, J. (2022). Deep active learning by leveraging training dynamics. Advances in Neural Information Processing Systems, 35, 25171-25184.

[4] Wen, Z., Pizarro, O., & Williams, S. (2023). NTKCPL: Active Learning on Top of Self-Supervised Model by Estimating True Coverage. arXiv preprint arXiv:2306.04099.

**Questions:**

1. On line 42, the citation to Ash. et al. seems broken.
2. On line 214, what is $v$?
3. On line 192, how is $\hat{\theta}_{\epsilon, x_i}$ defined as a function of $\epsilon$?
4. On line 250, shouldn't $C$ be a subset of the unlabeled examples instead of labeled examples?
5. The authors mention their estimation of the loss is unbiased and low bias at the same time. I think it's biased? Not sure why the authors call it unbiased.

---

### Official Review · Reviewer_o8n9 · 2024-11-04

**Soundness:** 3
**Presentation:** 3
**Contribution:** 2
**Rating:** 3
**Confidence:** 4

**Summary:**

This paper introduces an EER-based active learning approach for a low-budget setting. The proposed method comprises three main components: (i) label estimation through a surrogate model, (ii) model parameter update approximation using influence functions, and (iii) unbiased loss estimation. Additionally, the authors explore various techniques for batch-mode query selection. Extensive experiments on popular benchmark datasets suggest that this approach delivers superior performance over existing state-of-the-art AL methods.

**Strengths:**

The study addresses notable limitations in existing EER methods, potentially enhancing their applicability within deep learning frameworks. Of particular interest is the use of influence functions to approximate model parameter updates, thereby alleviating the need for repeated model retraining. The experiments are well-designed, covering a broad range of popular benchmark datasets. The results are robust and persuasive, demonstrating performance gains over existing active learning methods.

**Weaknesses:**

The main concern lies in the limited technical contributions of the methodology, as the primary techniques appear to have been adapted from existing studies. Besides, although influence functions offer an efficient approximation for parameter updates, they can be both memory and computationally intensive, especially with deep models. This may limit the practicality of applying the proposed method to deep models. A minor point is that the authors claim to use a surrogate model to estimate labels for unlabeled data. However, the implementation details suggest that the current model is merely assigning pseudo-labels rather than employing a distinct surrogate.

**Questions:**

1.	How does the running time of the proposed approach compare to other methods?
2.	Since this is an EER-based approach, the analysis of performance and computational efficiency between the proposed method and traditional EER algorithm would further substantiate the advantages claimed in the paper.
3.	In Section 3.3. the notion \bm{x} denotes only the data or represents the data-label pair?

---

### Note · Authors · 2024-12-04

**Comment:**

We thank the reviewers for their constructive reviews.

**Withdrawal Confirmation:**

I have read and agree with the venue's withdrawal policy on behalf of myself and my co-authors.